# A Little Is Enough:
# Circumventing Defenses For Distributed Learning

**Moran Baruch** [1]
moran.baruch@biu.ac.il

**Gilad Baruch** [1]
gilad.baruch@biu.ac.il

**Yoav Goldberg** [1] [2]
yogo@cs.biu.ac.il

[1] Dept. of Computer Science, Bar Ilan University, Israel
[2] The Allen Institute for Artificial Intelligence

## Abstract

Distributed learning is central for large-scale training of deep-learning models. However, it is exposed to a security threat in which Byzantine participants can interrupt or control the learning process. Previous attack models assume that the rogue participants (a) are omniscient (know the data of all other participants), and (b) introduce large changes to the parameters. Accordingly, most defense mechanisms make a similar assumption and attempt to use statistically robust methods to identify and discard values whose reported gradients are far from the population mean. We observe that if the empirical variance between the gradients of workers is high enough, an attacker could take advantage of this and launch a non-omniscient attack that operates within the population variance. We show that the variance is indeed high enough even for simple datasets such as MNIST, allowing an attack that is not only undetected by existing defenses, but also uses their power against them, causing those defense mechanisms to consistently select the byzantine workers while discarding legitimate ones. We demonstrate our attack method works not only for preventing convergence but also for repurposing of the model behavior ("backdooring"). We show that less than 25% of colluding workers are sufficient to degrade the accuracy of models trained on MNIST, CIFAR10 and CIFAR100 by 50%, as well as to introduce backdoors without hurting the accuracy for MNIST and CIFAR10 datasets, but with a degradation for CIFAR100.

## 1 Introduction

*Distributed Learning* has become a wide-spread framework for large-scale model training [1, 3, 10, 17, 18, 23, 31], in which a server is leveraging the compute power of many devices by aggregating local models trained on each of the devices.

A popular class of distributed learning algorithms is *Synchronous Stochastic Gradient Descent* (sync-SGD), using a single server (called *Parameter Server* - PS) and $n$ workers, also called *nodes* [17, 18]. In each round, each worker trains a local model on his or her device with a different chunk of the dataset, and shares the final gradients with the PS. The PS then aggregates the gradients of the different workers, and starts another round by sharing with the workers the resulting combined parameters to start another round. The structure of the network (number of layers, types, sizes etc.) is agreed between all workers beforehand.

While effective in a sterile environment, a major risk emerge with regards to the correctness of the learned model upon facing even a single *Byzantine* worker [5]. Such participants are not rigorously following the protocol either innocently, for example due to faulty communication, numerical error or crashed devices, or adversarially, in which the Byzantine output is well crafted to maximize its effect on the network.

We consider malicious Byzantine workers, where an attacker controls either the devices themselves, or even only the communication between the participants and the PS, for example by *Man-In-The-Middle (MITM)* attack. Both attacks and defenses have been explored in the literature [5, 11, 24, 28, 29].

In the very heart of distributed learning lies the assumption that the parameters of the trained network across the workers are independent and identically distributed (i.i.d.) [5, 9, 29]. This assumption allows the averaging of different models to yield a good estimator for the desired parameters, and is also the basis for most defense mechanisms, which try to recover the original mean after clearing away the byzantine values. Existing defenses claim to be resilient even when the attacker is omniscient [5, 11, 28], and can observe the data of all the workers. Likewise, most existing defenses for distributed learning [5, 11, 28, 29] work under the assumption that changes which are upper-bounded by an order of the variance of the correct workers cannot satisfy a malicious objective. This last assumption is advocated by the fact that SGD *better* converges with a little random noise [21, 25, 13]. Given that last assumption, those defenses use statistics-based methods to clear away the large changes and prevent attacks.

We show that this assumption is incorrect: the experimental variance between the different workers is high enough so that by carefully crafting byzantine values that are as far as possible from the correct ones, yet within the bounds of existing defenses, we are capable of defeating all state-of-the-art defenses and interfering with or gaining control over the training process. Moreover, while most (but not all) previous attacks focused on preventing the convergence of the training process, we demonstrate a wider range of attacks and support also introducing *backdoors* to the resulting model, which are samples that will produce the attacker's desired output, regardless of their true label. Lastly, by exploiting the i.i.d assumption we introduce a *non-omniscient* attack in which the attacker only has access to the data of the corrupted workers.[1]

**Contributions** We present a new approach for attacking distributed learning with this properties:

1. We provide a perturbation range in which the attacker can change the parameters without being detected **even in i.i.d. settings**.

2. Changes within this range are sufficient for both interfering with the learning process **and for backdooring the system**.

3. We propose the first non-trivial **non-omniscient** attack in i.i.d settings applicable for distributed learning, making the attack stronger and more practical.

4. The **same configuration** of the attack overcome all state-of-the-art statistics-based defenses.

## 2   Background

Distributed training is using the Synchronous SGD protocol, presented in Algorithm 1.

---

**Algorithm 1:** *Synchronous SGD*

---

**1** $P^1 \leftarrow$ Randomly initiate the parameters in the server.
**2** **for** *round $t \in [T]$* **do**
**3**   | The server sends $P^t$ to all $n$ workers.
**4**   | **for** *each worker $i \in [n]$* **do**
**5**   |   | Set $P^t$ as initial parameters and train locally using own data chunk.
**6**   |   | Return final parameters $p_i^{t+1}$ to the server.[2]
**7**   | $P^{t+1} \leftarrow AggregationRule(\{p_i^{t+1} : i \in [n]\})$
**8** **return** $P^t$ that maximized accuracy on the test set.

The attacker interferes the process at the time that maximizes its effect, that is between lines 5 and 6. During this time, the attacker can use the corrupted workers' parameters expressed in $p_i^{t+1}$, and replace them with whatever values it desires to send to the server. Attack methods differ in the way in which they set the parameter values, and defense methods attempt to identify corrupted parameters and discard them.

Algorithm 1 aggregates the workers values using averaging ($AggregationRule()$ in line 7). Some defense methods change this aggregation rule, as explained below.

**Notation.** All existing defenses are working on each round separately, so for the sake of readability we will discard the notation of the round ($^t$).

For the rest of the paper we will use the following notations:
$n$ is the total number of workers, $m$ is the number of malicious workers, and $d$ is the number of dimensions (parameters) of the model, $p_i$ is the vector of parameters trained by worker $i$, and $(p_i)_j$ is its $j$th dimension, and $\mathcal{P}$ is $\{p_i : i \in [n]\}$.

## 2.1 Malicious Objectives

**Convergence Prevention** is the attack which most of the existing literature for distributed learning with byzantine workers focuses on [5, 11, 28]. In this case, the attacker interferes with the process with the mere desire of obstructing the server from reaching good accuracy. In this type of attack the server is aware of the attack and, in a real world scenario, is likely to take actions to mitigate it, for example by actively blocking subsets of the workers and observing the effect on the training process.

**Backdooring** [4, 8, 19], is an attack in which the attacker manipulates the model at training time so that it will produce the attacker-chosen target at inference time. The backdoor can be either a single sample, e.g. falsely classifying a specific person as another, or it can be a class of samples, e.g. setting a specific pattern of pixels in an image to cause it to be classified maliciously. Bagdasaryan et al. [2] demonstrated a backdooring attack on *federated* learning by making the attacker optimize for a model with the backdoor while adding a term to the loss that keeps the new parameters close to the original ones. Their attack has the benefits of requiring only a few corrupted workers, as well as being non-omniscient. However, it does not work for distributed training: in federated learning each worker is using its own private data, coming from a different distribution, negating the i.i.d assumption [20, 14] and making the attack easier as it drops the ground under the fundamental assumption of all existing defenses for distributed learning. In [12], Fung et al. proposed a defense against backdoors in federated learning, but like the attack above it heavily relies on the non-i.i.d property of the data, which does not hold for distributed training.

A few defenses aimed at detecting backdoors were proposed [26, 22, 6, 27], but those defenses assume a single-server training in which the backdoor is injected in the training set for which the server has access to, so that by clustering or other techniques the backdoor samples can be found and removed from the training set. In contrast, in our settings, the server has no control over the samples which the workers adversely decide to train with, rendering those defenses inoperable. Finally, [24] demonstrate a method for circumventing backdooring attacks on distributed training. As discussed below, the method is a variant of the Trimmed Mean defense, which we successfully evade.

## 2.2 Coding-Based Defenses For Byzantine Workers

While all other defenses take a statistics-based approach, the authors of DRACO [7] suggest a coding-based defense. It is achieved by having the PS sending each chunk of data to multiple workers, and using majority to find the correct evaluation of each chunk. DRACO defense could be robust even against small changes, but it should be noted that it defends against a very limited number of Byzantine workers, not $O(n)$ such as Krum, MeanMed or Bulyan, to be detailed below. In real life, an attacker controlling a network component (e.g. a router or a switch) near the PS will be able to perform a Man-In-The-Middle attack while controlling more than a handful of nodes. In DRACO's settings, in case that the defender wishes to protect against up to $m = 10$ corrupted workers, each chunk needs to be calculated $r = 2m + 1 = 21$ times.

Ignoring the run-time concerns, it seems that only methods such as DRACO which force the output to be *identical* to results with no attack can insure resilience to attacks that require minor changes.

## 2.3 Statistics-Based Defenses For Byzantine Workers

The state-of-the-art defense for distributed learning is *Bulyan*. Bulyan utilizes a combination of two earlier methods - *Krum* and *Trimmed Mean*, to be explained first.

**Trimmed Mean.** This family of defenses, called *Mean-Around-Median* [28] or *Trimmed Mean* [29], change the aggregation rule of Algo 1 to a trimmed average, handling each dimension separately:

$$TrimmedMean(\mathcal{P}) = \left\{ v_j = \frac{1}{|U_j|} \sum_{i \in U_j} (p_i)_j : j \in [d] \right\}$$

Three variants exist, differing in the definition of $U_j$.

1. $U_j$ is the indices of top-$(n - m)$ values in $\{(p_1)_j, ..., (p_n)_j\}$ nearest to the median $\mu_j$ [28].

2. Same as the first variant only taking top-$(n - \mathbf{2}m)$ values [11].

3. $U_j$ is the indices of elements in the same vector $\{(p_1)_j, ..., (p_n)_j\}$ where the largest and smallest $m$ elements are removed, regardless of their distance from the median [29].

A defense method of [24] clusters each parameter into two clusters using 1-dimensional k-means, and if the distance between the clusters' centers exceeds a threshold, the values compounding the smaller cluster are discarded. This can be seen as a variant of the Trimmed Mean defense, because only the values of the larger cluster which must include the median will be averaged while the rest of the values will be discarded.

All variants are designed to defend against up to $\lceil \frac{n}{2} \rceil - 1$ corrupted workers, as this defenses depend on the assumption that the median is taken from the range of benign values.

The circumvention analysis and experiments are similar for all variants upon facing our attack, so we will consider only the second variant which is used in Bulyan below.

**Krum.** Suggested by Blanchard et al [5], *Krum* strives to find a single honest participant which is probably a good choice for the next round, discarding the data from the rest of the workers. The chosen worker is the one with parameters which are closest to another $n - m - 2$ workers, mathematically expressed by:

$$Krum(\mathcal{P}) = \left( p_i \mid \text{argmin}_{i \in [n]} \sum_{i \to j} \|p_i - p_j\|^2 \right)$$

Where $i \to j$ is the $n - m - 2$ nearest neighbors to $p_i$ in $P$, measured by Euclidean Distance.

Like TrimmedMean, Krum is designed to defend against up to $\lceil \frac{n}{2} \rceil - 1$ corrupted workers $(m)$. The intuition behind this method is that in normal distribution, the vector with average parameters in each dimension will be the closest to all the parameter vectors drawn from the same distribution. By considering only the distance to the closest $n - m - 2$ workers, sets of parameters which will differ significantly from the average vector are outliers and will be ignored. The malicious parameters, assumed to be far from the original parameters, will suffer from the high distance to at least one non-corrupted worker, which is expected to prevent it from being selected.

While Krum was proven to converge, in [11] the authors showed that convergence alone should not be the target, because the parameters may converge to an *ineffectual* model. In addition, as already noted in [11], due to the high dimensionality of the parameters, a malicious attacker can notably introduce a large change to a single parameter without a considerable impact on the L$^p$ norm (Euclidean distance), making the model ineffective.

Furthermore, The output of Krum's process is only one chosen worker, and all of its parameters are being used while the other workers are discarded. It is assumed that there exists such a worker for which all of the parameters are close to the desired mean in each dimension. In practice however, where the parameters are in very high dimensional space, even the best worker will have at least a few parameters which will reside far from the mean. To exploit this shortcoming, one can generate a set of parameters which will differ from the mean of each parameter by only a small amount. Those small changes will decrease the Euclidean Distance calculated by Krum, hence causing the malicious set to be selected.

**Bulyan.** El Mhamdi et al. [11], who suggested the above-mentioned attack on Krum, proposed a new defense that successfully oppose such an attack. They present a "meta"-aggregation rule, where another aggregation rule $\mathcal{A}$ is used as part of it. In the first part, Bulyan is using $\mathcal{A}$ iteratively

to create a *SelectionSet* of probably benign candidates, and then aggregates this set by the second variant of TrimmedMean. Bulyan combines methods working with $L^p$ norm that proved to converge, with the advantages of methods working on each dimension separately, such as TrimmedMean, overcoming Krum's disadvantage described above because TrimmedMean will not let the single malicious dimension slip.

Algorithm 2 describes the defense. It should be noted that on line 5, $n - \mathbf{4m}$ values are being averaged, which is $n' - \mathbf{2m}$ for $n' = |SelectionSet| = n - 2m$.

---

**Algorithm 2:** Bulyan Algorithm

**Input:** $\mathcal{A}, \mathcal{P}, n, m$
1   $SelectionSet \leftarrow \emptyset$
2   **while** $|SelectionSet| < n - 2m$ **do**
3     $p \leftarrow \mathcal{A}(\mathcal{P} \setminus SelectionSet)$
4     $SelectionSet \leftarrow SelectionSet \cup \{p\}$
5   **return** $TrimmedMean_{(2)}(SelectionSet$)

---

Unlike previous methods, Bulyan is designed to defend against only up to $\frac{n-3}{4}$ corrupted workers. Such number $m$ insures that the input for each run of $\mathcal{A}$ will have more than $2m$ workers as required, and there is also a majority of non-corrupted workers in the input to $TrimmedMean$. We follow the authors of Bulyan and use $\mathcal{A}$=Krum in the rest of the paper including the experiments.

**No Defense.** In the experiments section we will use the name *No Defense* for the basic method of averaging the parameters from all the workers, due to the lack of outliers rejection mechanism.

## 3   Our Attack

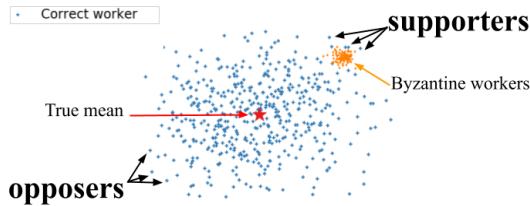

Figure 1: Illustration of the supporting and opposing correct workers.

As mentioned above, the research in the field of distributed learning assumes that the different parameters of all of the workers are i.i.d. and therefore expressed by normal distribution. We follow this assumption, hence in the rest of the paper the "units" of a change applied by the malicious opponent for attacking distributed learning models are standard deviations ($\sigma$).

Intuitively, the values of the correct workers are spread symmetrically around the mean, and at the extremes there are workers that push strongly to the direction we want ("supporters"), while a similar number of workers push to the opposite direction ("opposers"). By crafting values between the mean and the supporters, our byzantine values will be closer to the mean than the opposers are. This makes all existing defenses remove the non-byzantine opposer workers, while choosing workers (either byzantine or not) in the desired direction, shifting the mean (See Figure 1). The variance of the workers around the mean is large enough for this effect to be meaningful.

### 3.1   Exploited assumption

The attacks described in [5, 29, 11] share a similar assumption on the behaviour of the Byzantine workers - the attacker will choose parameters that are far away from the real mean of the parameters, for example by choosing parameters that are in the opposite direction of the gradient, in order to skew the real mean, and prevent the model convergence.

Therefore, the suggested defenses strive to prove that the selected set of parameters will lie within a ball centered at the real mean and with a radius which is a function of the correct workers. They

assume that if the selected parameters are close to a correct one, or there exist a correct worker which is even farther away, the model will produce accurate results.

In Krum [5] the authors acknowledge the fact that small changes will not be detected within a radius defined by $(\alpha, f)$, where the angle $\alpha$ depends on the ratio of the deviation over the gradients selected by their method to the real gradient, as a function of the proportion of corrupted workers $f$. They show that for any worker (potentially byzantine) $w$ selected by their method, there exists at least one **non-corrupted** worker $v$ for which $\|p_w - \vec{\mu}\| \leq \|p_v - \vec{\mu}\|$ where $\vec{\mu}$ is the mean of all $p_i$ for $i \in BenignWorkers$. The assumption exploited by our attack is that if such worker $v$ exist, choosing $p_w$ which is closer to the ideal target $\vec{\mu}$ will be at least as good as using parameters from a benign worker which is expected to produce good results. In practice, the Krum defense will be forced to either choose one of our byzantine workers, or a supporter.

In TrimmedMean [29] the authors bounded the error rate which their method guarantees as a function of the variance and skewness of the different workers. The larger the variance and skewness, the higher the error rate the attacker can achieve. They show an example: for any parameter vector $w$ and data point $(x, y) \in \mathbb{R}^d \times \mathbb{R}$, the variance $Var(\nabla f(w; x, y)) \leq V^2$. For convex model with quadratic loss, if the diameter of the parameter space is constant, $V = O(\sqrt{d})$ where $d$ is the dimension of the parameters. In such case the error rate is bounded by $\widetilde{\mathcal{O}}(\frac{m}{\sqrt{\ell}} + \frac{1}{\sqrt{\ell n}} + \frac{1}{\ell})$ where $m$ is the ratio of corrupted machines out of $n$ total, and $l$ is the number of data points on each machine.

As our experiments show (in Section 4.1), the high variance even on simple deep learning tasks, along with the high skewness induced by putting all the Byzantine workers on the same side in our attack below, is enough for the attacker to hide Byzantine workers within the variance of the correct workers, and still gain control over the training process.

We would like to induce directed *small* changes to many parameters, instead of large changes to a few. In the next section we will further explain how to choose such $p_{mal}$ for which normal distribution properties guarantee the existence of $s$ non-corrupted workers (which we called "supporters") for which $\|p_{mal} - \vec{\mu}\| \leq \|p_s - \vec{\mu}\|$. An illustration of our attack method is depicted in Figure 1.

In addition, the aforementioned defenses claim to protect against an **omniscient** attacker, i.e. who knows the data of all of the workers. We show that due to the normal distribution of the data, in case the attacker controls a representative portion of the workers, it is sufficient to have only the corrupted workers' data in order to estimate the distribution's mean and standard deviation, and manipulate the results accordingly. The variance of the estimation of the mean $\bar{X}$ is inversely proportional to the size of the set: $\bar{X} \sim \mathcal{N}(\mu, \frac{\sigma^2}{n})$. This observation enables our attack to work also for **non-omniscient** attacker, by estimating the properties of the entire population through the corrupted participants alone.

## 3.2  Perturbation Range

As mentioned before, the defense mechanisms are designed to discard values that are too far away from the mean. We thus seek a range in which we can deviate from the mean without being detected. Since normal distribution is symmetric, the same value $z^{max}$ will set the lower and upper bounds for the applicable changes around the mean.

What is the maximal change that can be applied by an attacker without being detected? In order to change the value produced by the aggregation rule, the attacker should control a majority of the workers. While existing defenses prevented it by claiming to support only up to $\lceil \frac{n}{2} \rceil - 1$ corrupted workers, the attacker can yet attain a majority by finding the minimal number $s$ of non-corrupted workers that are required as "supporters". The attacker will then use properties of the normal distribution, specifically the Cumulative Standard Normal Function $\phi(z)$, and look for the maximal value $z^{max}$ such that $s$ non-corrupted workers will reside farther away from the mean, hence preferring the selection of the byzantine parameters instead of the more distant correct mean. The exact steps for finding $z^{max}$ are shown in Algorithm 3, lines 1-4. By setting all corrupted workers to values in the range $(\mu - z^{max}\sigma, \mu + z^{max}\sigma)$, the defenses will not be able to differentiate the corrupted workers from the benign.

The probability for a worker to become a supporter can be considered as a Binomial with $p = 1 - \phi(z)$ for an attack factor $z$. Thus, the probability for finding $s$ supporters is $1 - I(\phi(z); n - m - s + 1, s)$ on each dimension independently where $I(z; a, b)$ is the *Regularized Incomplete Beta Function*.

**Algorithm 3:** Preventing Convergence Attack

**Input:** $\{p_i : i \in CorruptedWorkers\}, n, m$

1 $s \leftarrow \lfloor \frac{n}{2} + 1 \rfloor - m$

2 $z^{max} \leftarrow \max_z \left( \phi(z) < \frac{n-m-s}{n-m} \right)$

3 **for** $j \in [d]$ **do**

4     calculate mean ($\mu_j$) and std ($\sigma_j$)

5     $(p_{mal})_j \leftarrow \mu_j - z^{max} \cdot \sigma_j$

6 **for** $i \in CorruptedWorkers$ **do**

7     $p_i \leftarrow p_{mal}$

---

**Algorithm 4:** Backdoor Attack

**Input:** $\{p_i : i \in CorruptedWorkers\}, n, m$

1 Calculate $z^{max}$, $\mu$ and $\sigma$ as in Algo 3, lines 1-4.

2 Train the model parameters $\mathcal{V}$ with the backdoor, having initial parameters $\{\mu_j : j \in [d]\}$ .

3 **for** $j \in [d]$ **do**

4     $(p_{mal})_j \leftarrow clip(v_j,\ \mu_j - z_j^{max},\ \mu_j + z_j^{max})$

5 **for** $i \in CorruptedWorkers$ **do**

6     $p_i \leftarrow p_{mal}$

---

**TrimmedMean** is circumvented because by making the "supporters" prefer the malicious parameters, the attacker controls the median, and the final parameters after averaging the nearby parameters will be close to that. **Krum** indeed achieve its target of selecting a set of parameters for which there exist a benign worker which lies farther away, but with this parameters the selected parameters lie very close to the boundaries of this guarantee.

Since **Bulyan** is a combination of *Krum* and *TrimmedMean*, and since our attack circumvents both, it is reasonable to expect that it will circumvent Bulyan as well. Nevertheless, Bulyan claim to defend against only up to 25% of corrupted workers, and not 50% like Krum and TrimmedMean. At first glance it seems that the $z^{max}$ derived for $m = 25\%$ might not be sufficient, but it should be noted that the perturbation range calculated above is the possible input to *TrimmedMean*, for which $m$ can reach up to 50% of the workers in the $SelectionSet$ being aggregated in the second phase of Bulyan. Indeed, our approach is effective also against Bulyan.

The fact that the same set of parameters was used against all defenses is a strong advantage for this method: the attack will go unnoticed no matter which of the aforementioned statistics-based defenses the server decides to choose, again rendering our attack more practical.

### 3.3 Preventing Convergence

With the objective of forestalling convergence, the attacker will use the maximal value $z$ that will circumvent the defense. The attack flow is detailed in Algorithm 3.

**Example:** If the number of malicious workers is 24 out of a total of 50 workers, the attacker needs 2 "supporters" ($\lfloor \frac{50}{2} + 1 \rfloor - 24 = 2$) in order to have a majority and set the median. $\frac{50-24-2}{50-24} = 0.923$, and by looking at the z-table for the maximal $z$ for which $\phi(z) = 0.923$ we get $z^{max} = 1.43$. Finally, the attacker will set the value of all the malicious workers to $v = \mu_j - 1.43 \cdot \sigma_j$ for each parameter $j$ independently. Having enough workers with value lower than $v$, will set $v$ as the median.

### 3.4 Backdooring Attack

In section 3.2, we found a range in which the attacker can perturb the parameters without being detected, and in order to obstruct the convergence, the attacker maximized the change inside this range. For backdooring attack on the other hand, the attacker seeks the set of parameters within this range which will produce the desired label for the backdoor, while minimizing the impact on the functionality for benign inputs. To accomplish that, similar to [2], the attacker optimizes for the model with the backdoor while minimizing the distance from the original parameters. This is achieved by adding a term to the loss function weighted by a parameter $\alpha$ as: $\alpha\ell_{backdoor} + (1-\alpha)\ell_\Delta$ where $\ell_{backdoor}$ introduce the backdoor, and $\ell_\Delta$ is the MSE Loss between the new parameters and the original ones, keeping them close.

For $\alpha$ too large, the parameters will significantly differ from the original parameters, thus being discarded by the defense mechanisms. While for $\alpha$ too low the backdoor will not be introduced to the model. Hence, the attacker should use the minimal $\alpha$ which successfully introduce the backdoor in the model. This attack is detailed in Algorithm 4.

## 4 Experiments and Results

We provide two kinds of experiments: (1) empirically validating the claim regarding the variance between correct workers, and (2) validating the applicability of the methods by attacking real world networks. More experiments with different settings can be found in the supplementary material.

We experiment with attacking the following models, with and without the presence of the defenses. Following the experiments described in the state-of-the-art defenses [28, 29, 11], we consider simple architectures on the first two datasets: **MNIST** [16] and **CIFAR10** [15]. To strengthen our claims, we also experimented on the modern WideResNet architecture [30] on **CIFAR100**. The models architectures and hyper-parameters, can be found in the supplementary materials. The models were trained with $n = 51$ workers, out of which $m = 12 \approx 24\%$ were corrupted and non-omniscient.

### 4.1 Variance Between Correct Workers

In this experiment, we want to quantify the extent of the success rate of the proposed attack. We do so by considering *gradient sign-flipping*: In each round, the attacker checks the fraction of parameters for which the gradients can change direction (flip sign) with the given attack. This is applicable for a dimension $j$ if the size of the mean gradient ($|\mu_j|$) is smaller than the change that the attacker can introduce: $z\sigma_j$, for the $z$ described above and $\sigma_j$ the standard deviation of the gradients across the different workers.

The results in Figure 2 demonstrates the relation between the different variances and the size of the gradients, and their effect on the model's accuracy. The graphs show the evaluations on the 3 experimented tasks with $z \in \{\frac{1}{2}, 1, 2\}$. It is clear that many gradients can flip sign even with a change of only $\frac{1}{2}\sigma$. These results negate the assumption of most defenses (most explicitly by Krum) that the standard deviation is smaller than the gradients.

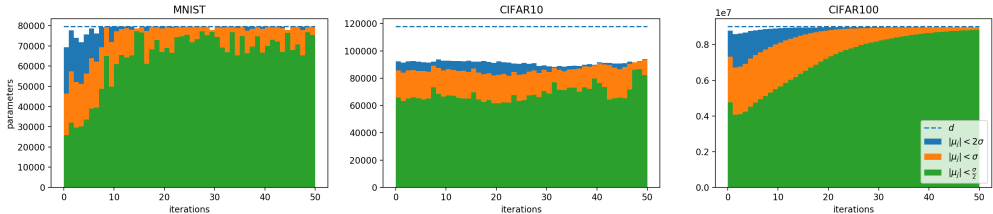

Figure 2: Parameters per iteration for which $|\mu_j| < z\sigma_j$, for $z \in \{\frac{1}{2}, 1, 2\}$ in the first 50 iterations.

### 4.2 Attack Objectives

Having established that the variance between correct workers is indeed high, we now demonstrate various kinds of attacks.

**Convergence Prevention.** In section 3.2 we showed that when the number of workers is 50, $z^{max}$ can be set to $1.43$. In the following experiments, we tried to change the parameters by up to $1.5\sigma$. In the supplementary materials we discuss about the needed $z$ for impacting the network convergence.

We applied our attack against all defenses, and examined their resilience on all models. Results can be found in Table 1. The reason for the low accuracy on CIFAR10 with no attack is the simple model we used as a reproduction of the defenses literature, and is consistent with the results described in those works. The $Krum$ defense performed worst, since our malicious set of parameters was selected even with only 24% of corrupted workers. *Bulyan* was affected more than *TrimmedMean*, because even though the malicious proportion was 24%, it can reach up to 48% of the *SelectionSet*, which is the proportion used by TrimmedMean in the second stage of Bulyan. *TrimmedMean* performed better than the previous two, because the malicious parameters were diluted by the averaging with many parameters coming from non corrupted workers.

Ironically but expected, the best defense strategy against this attack was the vanilla rule of averaging without outliers rejection. This is because the $1.5\sigma$ were averaged across all $n$ workers, 76% of which are not corrupted, so the overall shift in each iteration was $1.5 * 0.24 = 0.36\sigma$, which only have a

Table 1: **Convergence Prevention Results.** Maximal accuracy for $n = 51, m = 24\%, z = 1.5$.

| Model<br>Defense | MNIST | CIFAR10 | CIFAR100 |
|---|---|---|---|
| **No Attack** | 96.1 | 59.6 | 73.0 |
| **No Defense** | 91.1 | 42.2 | 63.1 |
| **Trimmed Mean** | 88.3 | 32.7 | 34.4 |
| **Krum** | 78.5 | 20.3 | 16.9 |
| **Bulyan** | 87.9 | 28.1 | 19.8 |

minor impact on the accuracy. It is clear however that the server cannot choose this aggregation rule because of the serious vulnerabilities it provokes. In case that circumventing *No Defense* is desired, the attacker can compose a hybrid attack, in which one worker is dedicated to attack *No Defense* with attacks detailed in earlier papers [5, 28], and the rest will be used for the attack proposed here.

**Backdooring.** As a result of the attacker's desire not to interrupt the convergence for benign inputs, low $\alpha$ and $z$ (both 0.2) were chosen. After each round the attacker trained the network with the backdoor for 5 rounds. We used MSE for $\ell_\Delta$ and set $\ell_{backdoor}$ to *cross entropy* like the one used for the original classification. On each round 1000 images were randomly sampled by the attacker, and their upper-left 5x5 pixels were set to the maximal intensity. All those samples were trained with $target = 0$. Testing was performed on a different subset.

Table 2 lists the results. MNIST perfectly learned the backdoor pattern with a minimal impact of 1% on benign inputs. For CIFAR10 the accuracy is worse than MNIST, with a degradation of 7 to 15% , but the accuracy drop for benign inputs is still reasonable and probably unsuspicious for an innocent server training for a new task without knowing the expected accuracy. Lastly, CIFAR100 was even less robust to the minimal changes for the backdoor, causing higher degradation to the accuracy.

It is interesting to see that *No Defense* was virtually resilient to this attack, with relatively minimal degradation on benign inputs and almost without mis-classifying samples with the backdoor pattern. However, on a different experiment on MNIST with higher $z$ and $\alpha$ (1 and 0.5 respectively), the opposite occur, where No Defense reached 95.6% for benign inputs and 100% on the backdoor, while other defenses did not perform as well on the benign inputs. Another option for circumventing *No Defense* is dedicating one corrupted worker for the case that *No Defense* is being used by the server, and use the rest of the corrupted workers for the defense-evading attack.

Table 2: **Backdoor pattern results.** The maximal accuracy with backdoor pattern attack. $n = 51$, $m = 24\%, z = \alpha = 0.2$. Results with no attack are also presented for comparison.

| | MNIST | | CIFAR10 | | CIFAR100 | |
|---|---|---|---|---|---|---|
| | Benign | Backdoor | Benign | Backdoor | Benign | Backdoor |
| **No Attack** | 96.1 | - | 59.6 | - | 73.0 | - |
| **No Defense** | 96.0 | 36.9 | 59.1 | 7.3 | 69.6 | 0.0 |
| **Trimmed Mean** | 95.3 | 100. | 55.6 | 80.7 | 69.7 | 100. |
| **Krum** | 95.2 | 100. | 52.5 | 95.1 | 52.8 | 99.8 |
| **Bulyan** | 95.3 | 99.9 | 51.9 | 84.3 | 54.9 | 92.9 |

## 5 Conclusions

We present a new attack paradigm, in which by applying limited changes to the reported parameters, a malicious opponent may **interfere with or backdoor** the process of *Distributed Learning*. Unlike previous attacks of byzantine workers, the attacker does not need to know the exact data of the non-corrupted workers (being **non-omniscient**), and it works even on i.i.d. settings, where the data is known to come from a specific distribution. The attack evades all state-of-the-art defenses based on robust aggregation, suggesting to use other approaches such as DRACO, albeit its run-time overhead.

## 6  Acknowledgements

This research was supported by the BIU Center for Research in Applied Cryptography and Cyber Security in conjunction with the Israel National Cyber Bureau in the Prime Minister's Office.

## Footnotes

[1]An exception to this line of defense is DRACO [7]. DRACO is taking a different approach on defending against byzantine workers, and uses coding based methods to insure the removal of byzantine values, including our attack. A longer discussion about this method and its real world applicability can be found in Section 2.2.

[2]In the original protocol, as in our experiments, the **gradients** are the ones to be published and aggregated.

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
