[Supplementary Material]

# Supplementary Materials

## 1 Illustration of possible attacks on a classifier

Figure 1: **Possible Malicious objectives. 1.** A normal scenario in which a benign image is classified correctly. **2.** The malicious opponent damaged the network functionality which now mis-classify legitimate inputs. **3.** A backdoor appear in the model, classifying this specific image as the attacker desire. **4.** The model produces the label **4** whenever a specific pattern (e.g. a square in the top left) is applied.

## 2 Additional Experiments

For our experiments we used PyTorch's [1] built in distribution package.

### 2.1 Models

For both MNIST and CIFAR10 datasets, we follow the model architecture of the paper introducing the state of the art Bulyan defense [3]. For MNIST, we use a multi-layer perceptron with 1 hidden layer, 784 dimensional input (flattened $28 \times 28$ pixel images), a 100-dimensional hidden layer with ReLU activation, and a 10-dimensional softmax output, trained with cross-entropy objective. By using this structure, $d$ equals almost 80k. We trained the model for 150 epochs with *batch size = 83*. When neither attack nor defense are applied, the model reaches an accuracy of $96.1\%$ on the test set.

For CIFAR10 we use a 7-layer CNN with the following layers: input of size 3072 ($32 \times 32 \times 3$); convolutional layer with kernel size: $3 \times 3$, 16 maps and 1 stride; max-pooling of size $3 \times 3$, a convolutional layer with kernel $4 \times 4$, 64 maps and 1 stride; max-pooling layer of size $4 \times 4$; two fully connected layers of size 384 and 192 respectively; and an output layer of size 10. We use ReLU activation on the hidden layer and softmax on the output, training the network for 400 epochs with a cross-entropy objective. In this setting $d \simeq 117K$. The maximal accuracy reached in this model with no corrupted workers is $59.6\%$, similar to the result obtained in [3] for the same structure.

We also evaluated our attacks on CIFAR100 dataset, using wide residual network (WRN) architecture presented in [6]. We adopted the existing implementation[1] to work in distributed settings. We used a widen factor of 4 with 40 layers.

In those settings the total number of parameters is $d = 8.97M$. The maximum accuracy achieved in this model with 51 correct workers is 73%.

In all models we set the *momentum* to be 0.9, and applied *fading learning rate* with initial learning rate of 0.1, and the fading rate was obtained according to [3] and [6] implementations. In the implementation of CIFAR100 we also used the same update rule, with fading rate of 2000. We added L2 regularization with weight $10^4$ and *cross entropy loss* for all models. The authors of [3] and [2] did not apply momentum to the training process. Therefore we also trained the models with the attacks and defenses without momentum, and achieved similar results. The training data was split between $n = 51 = 4 \cdot m + 3$ workers, with $m = 12$ corrupted workers.

**Single-Batch vs Entire Epoch** Two approaches exist when training a model in distributed settings. In the first approach, on each communication round the worker is reporting the gradients trained on a single batch alone [4], while in the other on each iteration the worker is using all of its data to report the gradients [5]. Those approaches introduce a trade-off between computation and communication.

The results reported by Bulyan were obtained with the single-batch setting, hence we chose that approach for MNIST and CIFAR10, in which we follow the network architecture and configuration in order to achieve a fair comparison. For CIFAR100 on the other hand, we could not adopt the same approach because even after 2000 iterations the training did not produce good results when using only a single batch in every iteration. Therefore for the experiments on CIFAR100 each worker used all of its local data in order to report the best gradients.

## 2.2 Proportion of malicious workers

Figure 2 shows the proportion of corrupted workers required to attack the training of CIFAR10 model. Since *Bulyan* designed to protect against up to 25% malicious workers, we tried to train the model with different $m$s up to that value, and tested how it affected the accuracy when the attacker changes all the parameters by $1\sigma$. One can see that Krum is sensitive even to a small amount of corrupted workers, thus even with $m = 5\%$ the accuracy drops by 33%. The graph shows that as expected, as the proportion of corrupted workers grows, the model's accuracy decreases, but even 10% can cause a major degradation with existing defenses other than not defending at all, which is not a realistic option.

Figure 2: Model accuracy with different proportion of corrupted workers ($m$) on CIFAR10. $z = 1$.

## 2.3 Required $z$

In order to learn how many standard deviations are required for impacting the network with the convergence attack, we trained all models in distributed learning settings four times, each time changing the parameters by $z = 0$ (no change), 0.5, 1 and 1.5 standard deviations. We did it for all the workers ($m = n$), on all parameters with no defense in the server.

As shown in Table 1, it is sufficient to change the parameters by $1.5\sigma$ or even $1\sigma$ away from the real average to substantially degrade the results. The table shows that degrading the accuracy of CIFAR10 and CIFAR100 is much easier than MNIST, which is expected given the difference in nature of the tasks: MNIST is a much simpler task, so less samples are required and the different workers will quickly agree on the correct gradient direction, limiting the change that can be applied. While for the

Table 1: The maximal accuracy of all models when changing all the parameters for all workers.

| Model \ $\sigma$ | 0 | 0.5 | 1 | 1.5 |
|---|---|---|---|---|
| MNIST | 96.1 | 89.0 | 82.4 | 77.8 |
| CIFAR10 | 59.6 | 28.4 | 20.9 | 17.5 |
| CIFAR100 | 73.0 | 32.2 | 16.0 | 8.5 |

harder, more realistic classification task of CIFAR10 or CIFAR100, the disagreement between the workers will be higher, which can be leveraged by the malicious opponent.

## 2.4 Sample Backdooring

While the paper presented the results when evaluating **Pattern** backdooring, here we show the results when trying to backdoor a specific sample. For this task, we chose each time one of the first 3 images from each training set and take their desired backdoored targets to be $(y + 1) \mod |Y|$ where $y$ is the original label and $|Y|$ is the number of classes. The results were averaged over the different images, and the reason we chose the first 3 is for reproduceability.

Results are presented in Table 2. Throughout the process, the network produced the malicious target for the backdoor sample in more than 95% of the time, including specifically the rounds where the maximal overall accuracy was achieved. As can be seen, for a simple task such as MNIST where the network has enough capacity, the network succeeded to incorporate the backdoor with less than 1% drop in the overall accuracy. The results are similar across the different defenses by cause of the low $z$ being used. For CIFAR10 however, where the convergence is difficult even without the backdoor for the given simple architecture, the impact is more visible and reaches up to 9% degradation.

Table 2: **Backdoor Sample Results.** The maximal accuracy of MNIST and CIFAR10 models with a backdoor sample. $n = 51, m = 24\%, z = \alpha = 0.2$. The results with no backdoor introduction are also presented for comparison.

| Defense \ Model | MNIST | CIFAR10 |
|---|---|---|
| No Attack | 96.1 | 59.6 |
| No Defense | 95.4 | 58.4 |
| Trimmed Mean | 95.4 | 57.9 |
| Krum | 95.3 | 54.4 |
| Bulyan | 95.3 | 54.2 |

## Footnotes

[1]https://github.com/xternalz/WideResNet-pytorch