[Reviews · NeurIPS 2019]

Reviewer 1



Originality: to play the devil's advocate, the key message of this paper is "outside their working hypothesis, mainstream defense mechanisms do not work", is not that somehow a tautology ? Clarity: the paper is fairly well written, my concern is more with what is missing, rather than the clarity of what is already there. Precisely, what is missing is a thorough analysis of what is really wrong in the aforementioned hypothesis. Significance: The main point of this paper is to show that the prerequisite hypothesis of previously published defenses might not always hold. In particular, the empirical variance of the correct workers might be too high to enable robust statistics methods to work. This paper would gain significance by elaborating more on that. I was surprised to see this question mentioned in the abstract but not that much elaborated in the main paper.

Reviewer 2



In general, I like the question this paper asked, i.e., whether or not it is necessary to impose a large deviation from the model parameters in order to attack distributed learning. Most of the research in Byzantine tolerant distributed learning, including Krum, Bulyan, and Trimmed Mean, uses some statistically "robust aggregation" instead of simple mean at the PS to mitigate the effects of adversaries. By the nature of robust statistics, all of those methods takes positive answer to the above question as granted, which serves as a cornerstone for their correctness. Thus, the fact that this paper gives a negative answer is inspiring and may force researchers to rethink about whether or not robust aggregation is enough for Byzantine tolerant machine learning. However, the author seems not aware of DRACO (listed below), which is very different from the baselines considered in this paper. L.Chen et al Draco: Byzantine-resilient distributed training via redundant gradients, ICML 2018. The key property of DRACO is that it ensures black-box convergence, i.e., it does not assume anything about the attack Byzantine workers use to achieve convergence. Thus, the "common assumption" is not made by DRACO, and it is NOT reasonable to claim " all existing defenses for distributed learning [ 5 , 10 , 27 , 28 ] work under the assumption". While this paper's idea is creative, it does not seem to be fully developed. The proposed attack is shown to break quite some existing defense methods empirically. Yet, does it break DRACO? If not, does it mean that DRACO is "the right solution" to the proposed attack? More discussion is in need. In addition, theoretically, when would the proposed attack break certain defense (depending on the parameters chosen, datasets, etc)? Is there a simple rule to decide how to break a certain defense? The experiments were also oversimplified. Only three models on small datasets with a fixed distributed system setup are far away from enough to validate an attack empirically. The main idea of this paper is clear, but the writing itself does not seem to be polished. A lot of typos exist, and some sentences are hard to understand. The use of math notations is messy, too. See below for a few examples. 1) Line 4, "are (a) omniscient (know the data of all other participants), and (b) introduce ...": "are (a)" should be "(a) are"; 2) Line 10, "high enough even for simple models such as MNIST": What model is MNIST? To the best of my knowledge, MNIST is a dataset, NOT a model (such issues exist in the experimental section as well); 3) Line 45, " Likewise, all existing defenses for distributed learning [ 5 , 10 , 27 , 28 ] work under the assumption that changes which are upper-bounded by an order of the variance of the correct workers cannot satisfy a malicious objective": This is hard to understand. Rephrasing is needed. 4) Algorithm 3, Line 2: It is not clear what objective the optimization problem maximizes. My guess is that the authors want to maximize z, which should be written as $\max z s.t. \phi(z) \leq \frac{n- m-s}{n-m}$; 5) Line 326, "state of the art" -> state-of-the-art. ================================================= Thank the authors for their explanation.

Reviewer 3



The paper presents a new attack on distributed learning systems. The threat model and targeted defenses are clearly explained, and the attack is explained nicely, with easy-to-follow intuition along with a more precise technical explanation. The properties of the proposed attack are strong: not only circumventing previously proposed defenses (and showing that some previous intuition is incorrect), but also being non-omniscient. The paper is a solid contribution. Questions and areas for improvement: The evaluation does not say whether the m=12 malicious workers are omniscient or not. If they are non-omniscient, it would be great to reiterate this. And if not, that seems like a serious issue: in that case, there would be no substantiation of the claim of a non-omniscient attack. Line 80 says "in this type of attack, the attacker does not gain any future benefit from the intervention." This may not be true without additional context. For example, consider a scenario where a company attacks the distributed training of a competitor: in this case, there is clear benefit to performing the attack. It would be interesting to see more discussion about where such attacks/defenses may be applicable in the real world, along with further elaboration of the threat model. In current practical uses of distributed training, are all nodes trusted? What future systems might be susceptible to such attacks? Lines 215--221 discuss the situation when the attacker is not omniscient. In particular, the paper says that when "the attacker controls a representative portion of the workers, it is sufficient to have only the workers' data in order to estimate the distribution's mean and standard deviation ..." What is a representative portion? It would be great to briefly summarize the threat model in the abstract (e.g. "percentage of workers are corrupted and collude to attack the system, but are non-omniscient and don't have access to other nodes' data"). Nits: - Line 164: "(n-3)/4" should presumably be "n/4"? - Figure 1 is unreadable in grayscale (e.g. when printed); consider using different symbols rather than just colors (and perhaps make it colorblind-friendly as well) - Line 253: "the attack will go unnoticed no matter which defense the server decides to choose" -- this only applies for the known, particular defenses examined in this paper

[Author Response · NeurIPS 2019]

We thank the reviewers for taking their time to carefully review our paper, for considering our attack method to be novel
and interesting, and for suggesting improvements.

Our main point is indeed showing that existing robust-statistics methods are not enough in many real-world use cases.
We find significance in identifying assumptions that do not hold in the real world. Rev2 described it well in their review.

As a method to foretell the success rate of the attack, the attacker can check the ratio of parameters for which the
gradients can change direction with the given attack. This is applicable for a dimension $j$ when the size of the mean
gradient ($|\mu_j|$) is smaller than the change that the attacker can introduce: $z\sigma_j$, for the $z$ described in the paper and $\sigma_j$ the
standard deviation of the gradients across the different workers. Figure 1 shows the calculations on the 3 experimented
tasks with $z \in \{\frac{1}{2}, 1, 2\}$. It is clear that many gradients can change direction even with a change of only $\frac{1}{2}\sigma$. These
results negate the assumption of most defenses (most explicitly by Krum) that the standard deviation is smaller than the
gradient itself. We will extend this analysis including discussion on backdooring for the final version.

Figure 1: Number of parameters per iteration where $|\mu_j| < z\sigma_j$, for $z \in \{\frac{1}{2}, 1, 2\}$ in the first 50 iterations.

We would like to thank Rev2 for bringing DRACO to our attention. We were not familiar with that work, which we
surely should and will discuss for the camera-ready version, as it should be resilient to our attack. DRACO takes a
coding-based approach instead of the robust-aggregation approach used by the defenses referred to in our paper. It
is achieved by having the PS sending each chunk of data to multiple workers, and using majority to find the correct
evaluation of each chunk. In that regards we have 2 remarks:

**a)** DRACO defends against a very limited number of byzantine workers, not $\mathcal{O}(n)$ such as Krum, TrimmedMean or
Bulyan. For our experiments where $m = 12$ workers were corrupted, each chunk needs to be calculated $r = m * 2 + 1 =$
25 times. This implies that training process that should have taken 2 days without defense will take almost 2 months
with DRACO. We disagree with the authors of this defense that only a few workers can be corrupted in real life. For
example, an attacker controlling a network component (e.g. a router or a switch) near the PS will be able to perform a
Man-In-The-Middle attack by adopting our method while controlling more than a handful of nodes.

**b)** DRACO does not prove its superiority over robust-statistics methods in the face of a specific attack. We show that
DRACO's contribution is more significant than run-time improvement, because only methods that force the results to be
**identical** to results without Byzantine workers such as DRACO will be resilient to attacks that require minimal changes.
Consecutively, we will limit our claims for overcoming existing defenses based on robust-statistics methods only.

**Models and datasets chosen:** First, for MNIST and CIFAR10 we followed the models and hyper-parameters selected
by the authors of Bulyan (the baseline defense we overcome) for a fair comparison. We added CIFAR100 with
WideResNet architecture in order to test our attack on a more realistic task. Our results show that our attack works on all
ranges of tasks, from the simplest (MNIST with 2 fully connected layers), through quite simple ConvNet for CIFAR10,
to the much more complex WideResNet architecture on CIFAR100 dataset. We find it interesting to present results on
MNIST, showing that the variances are high enough even for such a simple task. As of the request for more experiments
with different settings, we would like to point the reviewers to the supplementary material, where we evaluated our
methods on different number of Byzantine nodes (Figure 2) and different shifting factors (Table 1). We can also add
results which were not included initially showing that the results are similar for different number of overall workers ($n$).

**Non-omniscience:** All $m$ corrupted workers report **only their private data** to the attacker, meaning that the attacker is
indeed non-omniscient. We will reiterate it for the final version.

**Real-World Relevance:** Deep Learning became a research field which involves tremendous amount of money, and
various players are motivated to prevent a company from reaching high accuracy on some task, or backdooring their
model. This includes hacking into many training nodes or a central networking component as mentioned above. Another
trend amplifying this risk is the rise of start-ups allowing people to get paid for hosting training tasks on their private
GPUs in a distributed fashion while idle. In such model, the workers obviously cannot be trusted.

We will revise all cases which are pointed by the reviewers to be misleading, odd or unclear. We will also fix typos.

[Meta-Review · NeurIPS 2019]

The paper provides a new strong attack against robust byzantine ML training algorithms. This attack seems to be effective across a wide range of settings, and hence is a useful contribution to the related byzantine ML literature. One strong suggestion made during the rebuttal discussion is that the authors need to clarify the setting that they operate on, eg some methods they attack may still be robust on their original setting, but are not against the proposed scheme. That is the threat model needs to be clarified in detail.